# A high-throughput method to identify trans-activation domains within transcription factor sequences

Cosmas D Arnold[1,†], Filip Nemčko[1,†], Ashley R Woodfin[1,†], Sebastian Wienerroither[1,†], Anna Vlasova[1,†], Alexander Schleiffer[1,2], Michaela Pagani[1], Martina Rath[1] & Alexander Stark[1,3,*]

## Abstract

Even though transcription factors (TFs) are central players of gene regulation and have been extensively studied, their regulatory *trans*-activation domains (tADs) often remain unknown and a systematic functional characterization of tADs is lacking. Here, we present a novel high-throughput approach *tAD-seq* to functionally test thousands of candidate tADs from different TFs in parallel. The tADs we identify by pooled screening validate in individual luciferase assays, whereas neutral regions do not. Interestingly, the tADs are found at arbitrary positions within the TF sequences and can contain amino acid (e.g., glutamine) repeat regions or overlap structured domains, including helix–loop–helix domains that are typically annotated as DNA-binding. We also identified tADs in the non-native reading frames, confirming that random sequences can function as tADs, albeit weakly. The identification of tADs as short protein sequences sufficient for transcription activation will enable the systematic study of TF function, which—particularly for TFs of different transcription activating functionalities—is still poorly understood.

**Keywords** glutamine-rich regions; high-throughput functional screen; trans-activation domain; transcription; transcription factor
**Subject Categories** Methods & Resources; Systems & Computational Biology; Transcription
**The EMBO Journal (2018) 37: e98896**

## Introduction

The regulation of gene expression is central to development and cellular differentiation (Levine & Tjian, 2003), and erroneous gene expression is associated with many diseases including cancer (Herz *et al*, 2014; Bhagwat & Vakoc, 2015). At the transcriptional level, gene expression is determined by genomic *cis*-regulatory promoter and enhancers (Banerji *et al*, 1981), elements that bind regulatory transcription factor (TF) proteins (Shlyueva *et al*, 2014). Promoter- and enhancer-bound transcription factors, typically via cofactor (COF) proteins, mediate RNA Polymerase II (Pol II) recruitment and activation at core promoters (Spitz & Furlong, 2012; Reiter *et al*, 2017). TFs therefore assume a key position in transcriptional regulation, linking *cis*-regulatory DNA sequences to the regulatory COFs or the pre-initiation complex (PIC) including Pol II; in other words, they read and interpret *cis*-regulatory DNA sequence information. The central role of TFs in gene expression is highlighted by the fact that their activity can determine different cell types and transitions between them, including the dedifferentiation (also called *reprogramming*) of embryonic fibroblasts into pluripotent stem cells by the expression of Oct3/4, Sox2, c-Myc, and Klf4 (Takahashi & Yamanaka, 2006) or the conversion of fibroblasts into myoblasts by the expression of MyoD (Davis *et al*, 1987). In addition, deregulated or mutated TFs are causal to many diseases including cancer that is associated with the proto-oncogene c-Myc and the tumor suppressor p53, both of which are TFs (Scian *et al*, 2004).

A prototypical TF possesses two different functionalities: (i) the recognition and sequence-specific binding to short DNA sequence motifs; and (ii) the trans-activation of transcription, typically via the recruitment of COFs (Chrivia *et al*, 1993; Conaway & Conaway, 2013; Zabidi & Stark, 2016) or the direct interaction with the PIC (Choy & Green, 1993). These two functions are distinct and typically mediated by two different protein domains (Fig 1A): a DNA-binding domain (DBD) and a trans-activation domain [tAD, often also *TAD*, which we avoid given the frequent use of TAD for *topologically associating domains*, meaning self-associating chromosomal neighborhoods (Dixon *et al*, 2012; Nora *et al*, 2012)]. This functional separation can be demonstrated by splitting the two domains experimentally and assessing their trans-activating functions individually (Keegan *et al*, 1986; Fig 1A): The DBD of the yeast TF Gal4 (Gal4-DBD) alone is able to bind DNA but cannot trans-activate, while tADs are sufficient for trans-activation, e.g., when fused to the DBDs of other TFs (Brent & Ptashne, 1985). The ability to

1 Research Institute of Molecular Pathology (IMP), Vienna BioCenter (VBC), Vienna, Austria
2 Institute of Molecular Biotechnology (IMBA), Vienna BioCenter (VBC), Vienna, Austria
3 Medical University of Vienna, Vienna BioCenter (VBC), Vienna, Austria
*Corresponding author. Tel: +43 1 79730 3380; E-mail: stark@starklab.org
†These authors contributed equally to this work

trans-activate when fused to heterologous DBDs such as Gal4-DBD has been used to test candidate tADs, e.g., for GCN4, MTF-1, and other TFs (Figs 1A and 2; Hope & Struhl, 1986; Ma & Ptashne, 1987b; Günther et al, 2012), and to demonstrate that tAD functionality is deeply conserved between eukaryotes: the yeast TF Gal4, for example, functions in fly (Fischer et al, 1988), humans (Kakidani & Ptashne, 1988), and plant cells (Ma et al, 1988).

While DBDs are typically well-structured and display clear sequence similarities between orthologous and paralogous TFs (the classification of TFs into different families such as the homeobox, helix–turn–helix, or leucine zipper TFs (Luscombe et al, 2000) is, for example, based on these similarities), tADs are typically unstructured (Dyson & Wright, 2005) and share little sequence similarity. In fact, sequence similarities between tADs of different TFs are often restricted to the presence of short peptide motifs or characteristic amino acid (AA) compositions, including acidic-rich, glutamine-rich, or proline-rich domains (Gerber et al, 1994; Piskacek et al, 2007), and in contrast to the deeply conserved tAD functionality in transcription activation (Fischer et al, 1988; Kakidani & Ptashne, 1988; Ma et al, 1988), tAD sequences are not well conserved between orthologous TFs.

This lack of extended sequence similarity and evolutionary conservation makes the computational identification of tADs difficult, typically requiring time-consuming experimental testing of individual candidate fragments. Therefore, the tADs for most TFs remain unknown and a functional annotation of regulatory domains for different TFs is lacking, which might have impacted our understanding of the mechanisms by which different TFs activate transcription, a question of central importance that has remained unanswered (Erkina & Erkine, 2016).

Here, we present a novel high-throughput approach *trans-activation domain sequencing* or *tAD-seq* to functionally identify and characterize tADs within complex candidate libraries. The tADs we identify validate in individual luciferase assays, whereas neutral regions do not, localize to different positions within TF sequences and display diverse sequence signatures, including poly-glutamine repeats or structured DBD-like domains. We also identify tADs in non-native reading frames, confirming previous reports that arbitrarily selected sequences can function as tADs (Ma & Ptashne, 1987b; Abedi et al, 2001), albeit weakly. The systematic identification of tADs with methods like tAD-seq should uncover the requirements of tAD function, which—particularly for TFs of distinct transcription activating functionalities (Stampfel et al, 2015)—are still elusive, yet are key to reveal how TFs employ tADs to regulate transcription.

## Results

### tAD-seq—pooled screening of tAD candidate sequences

We conceived a high-throughput screening method *tAD-seq* to identify tADs directly by their trans-activating functions. tAD-seq is based on the established property of tADs to activate transcription even outside their endogenous sequence contexts, e.g., when fused to heterologous DBDs such as the Gal4-DBD (Ma & Ptashne, 1987b; Fig 1A). Our strategy multiplexes previous approaches that tested individual candidate tADs one by one in reporter assays (e.g., Keegan et al, 1986; Ma & Ptashne, 1987b; Seipel et al, 1992) and

screens based on selectable markers in yeast (Ma & Ptashne, 1987b; Abedi et al, 2001). Similar to these previous approaches, tAD-seq should be able to identify tADs that correspond to continuous peptides but not multi-partite tADs (e.g., Herbig et al, 2010).

tAD-seq enables the screening of complex pools of candidate fragments highly parallelized in a single experiment (Fig 1B) by the combination of (i) candidate recruitment to the promoter region of a selectable reporter gene (GFP under the control of Gal4-binding motifs (upstream activating sequences; UASs), hereafter called "UAS-GFP"); (ii) selection via fluorescent-activated cell sorting (FACS); and (iii) candidate mRNA quantification by next-generation sequencing (NGS). In brief, we clone candidate fragments downstream of a Gal4-DBD open reading frame (ORF) and upstream of a poly-adenylation site to create a complex library of expression clones for Gal4-candidate fusion proteins, driven by a constitutively active promoter. We co-transfect the Gal4-candidate library and the UAS-GFP reporter plasmids into *Drosophila* S2 cells, and separate GFP-positive (GFP$^+$) and GFP-negative (GFP$^-$) cells by FACS. Assuming that GFP$^+$ cells are enriched in candidates with tAD function, we determine the relative abundance of Gal4-candidate mRNA levels by NGS in GFP$^+$ vs. GFP$^-$ cells and identify tADs as regions that are enriched in GFP$^+$ cells (Fig 1B).

### GFP-FACS can enrich transcription activating factors

As a proof of concept, we first assessed whether the combined use of a UAS-GFP reporter, FACS, and mRNA quantification as described above is able to separate regulatory proteins based on their ability to activate transcription. We transfected a mix of expression constructs for ten full-length TF or COF proteins fused to the Gal4-DBD, which we previously tested to be activating, neutral, or repressive (Stampfel et al, 2015), together with UAS-GFP reporter plasmids into *Drosophila* S2 cells. We separated GFP$^+$ and GFP$^-$ cells by FACS and measured the relative mRNA levels for each Gal4-DBD fusion by RT–qPCR (Fig 1C). As expected, the two strongest activators (Sox14 and Labial [lab]) were enriched in GFP$^+$ cells, while the strongest repressors (Groucho [gro], Mirror [mirr]) were most strongly enriched in GFP$^-$ cells. Overall, the ability to activate or repress transcription as measured in individual luciferase assays (Stampfel et al, 2015) correlated well with the differential distribution between GFP$^+$ and GFP$^-$ cells in the pooled analysis (Pearson correlation coefficient (PCC) = 0.89; Fig 1C), suggesting that cellular GFP levels can be used to separate activators and repressors from a pool of candidates and that the above setup might allow the identification of tADs within complex candidate libraries.

### tAD-seq recovers the known tAD of MTF-1

To assess whether the setup above is also able to enrich tADs from within highly complex candidate pools, we performed a proof-of-principle screen, asking whether the known tAD of MTF-1 (Günther et al, 2012) could be recovered (Fig 2A). We chose 180 TFs (Table EV1) of which 68 activated transcription more than twofold and 32, including MTF-1, activated transcription more than fivefold (all others were neutral [89] or repressive [23]; Stampfel et al, 2015), randomly sheared their intronless cDNA-derived coding sequences (CDS) by sonication, and selected roughly 250-bp-long fragments that we cloned downstream of the Gal4-DBD open reading frame,

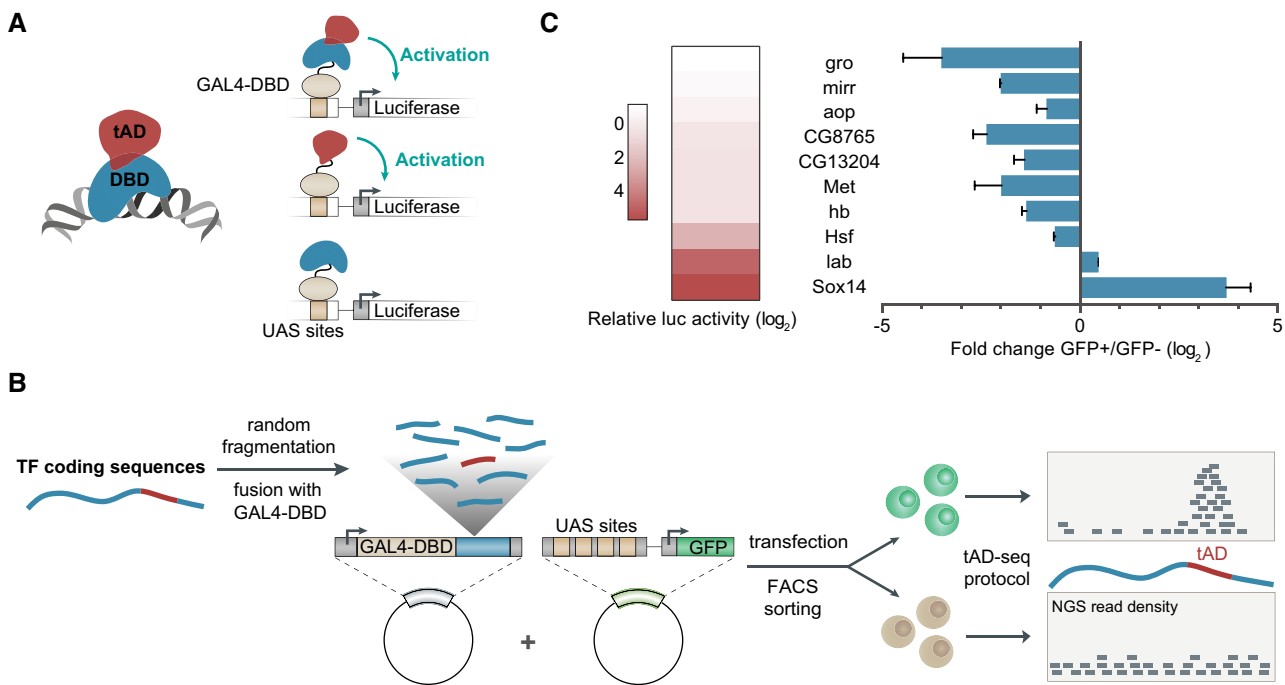

**Figure 1. Trans-activation domains (tADs) and their identification by tAD-seq.**

A   TFs are typically modular with two distinct functionalities—they bind to specific DNA sequences via their DNA-binding domains (DBDs, blue), and trans-activate transcription via trans-activation domains (tADs, red). tADs but not DBDs are sufficient to activate transcription when recruited to the promoter of a reporter gene (e.g., luciferase) via a heterologous DBD, here the Gal4-DBD.

B   Detailed schematic overview of the tAD-seq workflow, including Gal4-DBD-candidate library cloning (tAD-seq library), co-transfection of library and 4xUAS-GFP reporter plasmids, separation of GFP+ (tAD enriched) and GFP− cells by FACS, and NGS-based tAD identification by quantification of Gal4-DBD-candidate transcripts in GFP+ vs. GFP− cells.

C   TF-mRNA enrichment in GFP+ vs. GFP− cells reflects the TFs activating and repressing functionalities. Left: heat map depicting transcription activating and repressing functions (shades of red, see color legend) of the ten indicated TFs tested individually by recruitment to 4xUAS-luciferase reporters (data from Stampfel *et al*, 2015). Right: bar plot indicating relative distribution of TF transcripts between GFP+ and GFP− cells as measured by RT–qPCR (n = 3, error bars: s.d.) after transfecting a pool of ten TFs into S2 cells and separation of GFP+ and GFP− cells by FACS.

which is under the control of a strong promoter (derived from *ubiquitin-63E*), and upstream of a poly-adenylation site, using a protocol we recently established for the cloning of genome-wide candidate fragment libraries (Arnold *et al*, 2013). This yielded a complex candidate library with at least 538,856 distinct candidate fragments as determined by non-exhaustive NGS.

We transfected the candidate library together with the UAS-GFP reporter plasmid into S2 cells, selected GFP+ and GFP− cells by FACS, and isolated poly-A+ mRNAs from both cell pools. We reverse-transcribed Gal4-DBD-candidate mRNAs by a reporter-specific reverse transcription (RT) primer, which also contained a 10 nucleotide (nt) random barcode as a unique molecular identifier (UMI, e.g., Klein *et al*, 2015) that allows the counting of individual Gal4-DBD-candidate mRNAs. We then selectively amplified Gal4-DBD-candidate cDNAs by a nested PCR approach similar to the one used for STARR-seq (Arnold *et al*, 2013), and sequenced the resulting cDNA libraries from GFP+ and GFP− cells by paired-end NGS. We mapped the paired-end reads to the full-length TF coding sequences, used the UMIs to count individual candidate mRNAs, and scored their abundance in a position- and reading frame-specific manner. The resulting position-specific coverage is highly similar between two biological replicates (independent experiments; PCC = 0.89 for GFP+ cells and PCC = 0.94 for GFP− cells); we pooled the replicates

and used the normalized ratio between the position-specific coverage in GFP+ and GFP− cells, i.e., the enrichment in GFP+ cells to score trans-activating function characteristic of tADs.

For MTF-1, we observed a strong tAD-seq enrichment (14.5-fold; hypergeometric *P*-value = 0; FDR = 0) toward the middle of MTF-1's CDS, overlapping with the known tAD (Günther *et al*, 2012; Fig 2A). This strong enrichment was specific to the native (+1) reading frame (Fig 2A), even though random sequences can give rise to functional tADs (see below and Ma & Ptashne, 1987b; Abedi *et al*, 2001). When we tested the identified MTF-1 tAD individually in luciferase assays vs. a flanking region that tAD-seq predicted to be functionally neutral, we confirmed the strong activating function of the MTF-1 tAD (Günther *et al*, 2012) and the neutrality of the control region (Fig 2B). Together, this demonstrates that tAD-seq can recover functional tADs from a complex pool of candidate fragments.

### Identification of novel tADs by tAD-seq

To identify tADs *de novo*, we assessed the enrichment of candidate fragments in the native (+1) reading frame for all other TFs in the library (enrichment ≥ 1.5; hypergeometric *P*-value ≤ 1 × $10^{-7}$), which revealed 53 tADs in 49 TFs with enrichments up to 129-fold (all had FDRs ≤ 1.46 × $10^{-7}$; Table EV2; Appendix Fig S1). The TFs

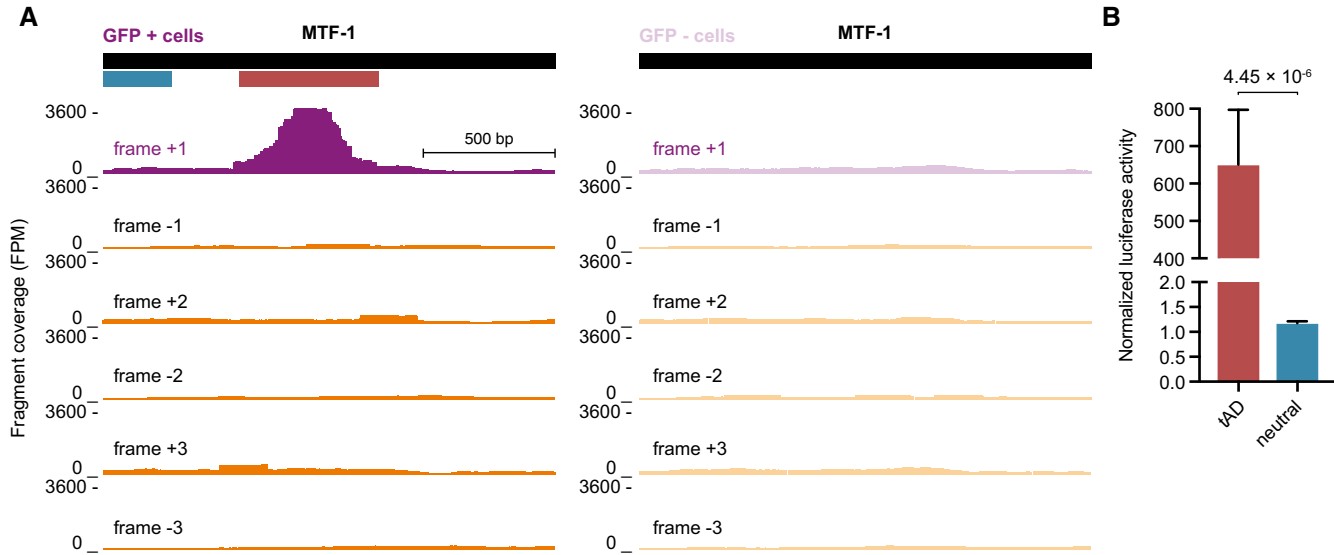

**Figure 2. tAD-seq recovers the known tAD of MTF-1 from a complex pool of candidates.**

A  UCSC Genome Browser (GB) screenshots (dedicated genome containing only TF CDSs and flanking plasmid backbone sequence, see Materials and Methods) displaying candidate fragment coverage for GFP⁺ cells (left) and GFP⁻ cells (right) in a reading frame-specific manner (+1 frame, purple and non-native frames, orange). The black bar on top indicates the full-length CDS of MTF-1. The colored bars below indicate the regions individually tested in luciferase assays (red = tAD; blue = neutral control region). High coverage is only observed in the native (+1) frame at the position of the known tAD.

B  Relative luciferase activity of sequences that overlap the MTF-1 tAD (red) or a neutral region (blue, see also A). Shown are the normalized luciferase activities for tAD candidates and neutral fragments (Gal4-DBD-candidate) normalized to a negative control (Gal4-DBD-GFP; $n = 4$, error bars: s.d., $P$-value: two-sided Student's $t$-test vs. neutral region; FPM fragments per million).

---

included Bteb2 for which the predicted tAD is near the N-terminus (Fig 3A), consistent with the location of the tAD of the human Bteb2 ortholog (Cao *et al*, 2010), and Clock (Clk) for which the predicted tAD is near the C-terminus, consistent with the location of the known tAD and the circadian rhythm defects in flies lacking this region (Allada *et al*, 1998). For eight TFs, we used luciferase assays to individually assess the trans-activation functions of the identified tADs and neutral control regions, which confirmed the tAD-seq results in each case (Figs 2 and 3A–G; and Table EV3). All in all, we tested 21 identified tADs and eight neutral control regions individually in luciferase assays. Overall, 67% (14 of 21) tAD candidates activated transcription in luciferase assays at least twofold (two-tailed Student's $t$-test $P < 0.05$) and 81% (17 of 21) at least 1.5-fold ($P < 0.05$; except for one tAD with $P = 0.058$), in contrast to none of the eight neutral regions (Fig 3H and Table EV3). This establishes tAD-seq as a high-throughput method to assess the activating potential of fragmented protein-coding sequences, allowing the unbiased testing and identification of tADs.

**Peptides translated from non-native reading frames function as weak tADs**

An unbiased screen based directly on tAD function should allow the discovery of random protein sequences with trans-activating functions (Ma & Ptashne, 1987b; Abedi *et al*, 2001). Indeed, when we scored tAD-seq enrichments for fragments cloned in the five non-native reading frames (enrichment ≥ 1.5; hypergeometric $P$-value ≤ $1 \times 10^{-7}$), we found 103 putative "out-of-frame" tADs with enrichments up to eightfold (Table EV4; Appendix Fig S2). For example, fragments from the repressive TF engrailed (en; Jaynes &

O'Farrell, 1991; Han & Manley, 1993; Stampfel *et al*, 2015) cloned in reading frame +3 were enriched in tAD-seq (Fig 4A; no tAD was detected in any other frame including frame +1, consistent with the repressive function of en) and indeed activated luciferase expression when tested individually (2.1-fold, Student's $t$-test, $P = 2.8 \times 10^{-3}$; Fig 4B). Similarly, tADs identified in reading frame +2 and reading frame +3 of bobby sox (bbx) and Sequoia (seq), respectively, significantly, yet weakly, activated transcription (2.1-fold and 1.9-fold; $P \leq 1.7 \times 10^{-4}$; Fig 4C–E and Table EV3). Overall, we tested seven out-of-frame tADs in single-candidate luciferase assays and found three (43%) to activate luciferase expression more than twofold relative to GFP ($P < 0.05$) and six (86%) more than 1.5-fold ($P < 0.05$), however at most 2.8-fold (Table EV3). Taken together, the analysis of the five non-native reading frames confirms previous reports that random peptide sequences can function as tADs (Ma & Ptashne, 1987b; Abedi *et al*, 2001) and that they are weak compared to strong native tADs (Ma & Ptashne, 1987b).

**tADs are at arbitrary positions along the TF sequences and can overlap structured DBDs**

The 53 native tADs are at different positions within the TFs' coding sequences, toward the N-terminus (e.g., Bteb2, Fig 3A), the middle (e.g., MTF-1, Fig 2A), or the C-terminus (e.g., CG14451, Fig 3C). Together with the lack of a clear sequence signature, this flexible location has made it difficult to predict tADs computationally. Interestingly though, while tADs are typically outside globular domains (thus the relative lack of protein family signatures), 12 tADs overlap structured protein domains as defined by the Pfam protein family database. For example, the tADs in *Hormone-receptor-like in 39* and

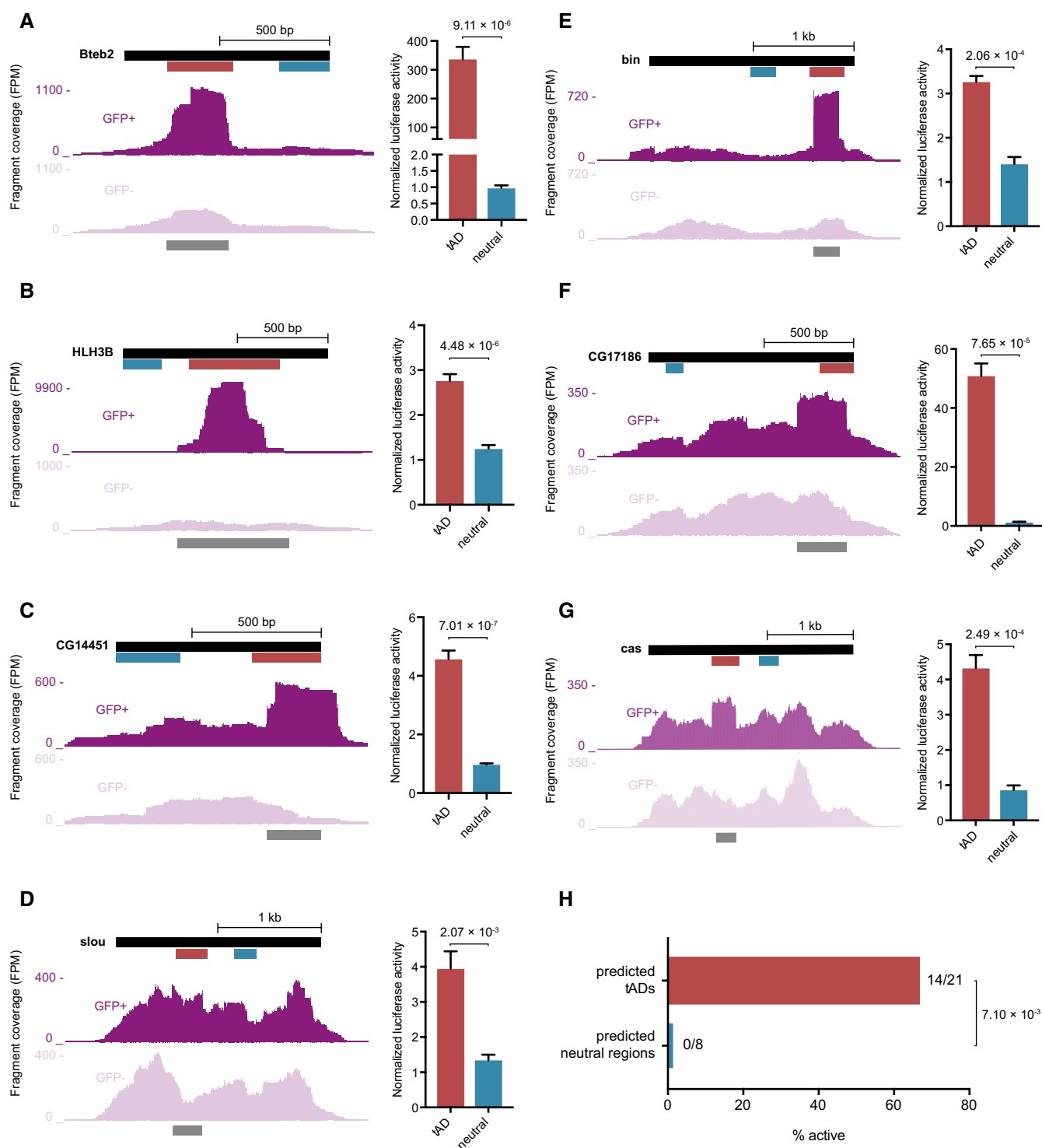

**Figure 3.  tAD-seq identifies novel tADs from a complex pool of candidates.**

A–G  Candidate fragment coverage (+1 frame) from GFP⁺ (top) and GFP⁻ (bottom) cells for Bteb2 (A), HLH3B (B), CG14451 (C), slou (D), bin (E), CG17186 (F), and cas (G). The dark gray bar at the bottom indicates the called tAD region and the red and blue bars the positions of the tAD candidate and the neutral control region, respectively, tested in luciferase assays. Normalized luciferase activities (normalized to GFP control) of tAD candidate and neutral control are shown on the right ($n = 4$ for Bteb2, HLH3B, and CG14451; $n = 3$ for slou, bin, CG17186, and cas; error bars: s.d., $P$-value: two-sided Student's $t$-test vs. neutral region).

H     Summary of individual tAD activity tests by luciferase assays for candidate tADs and neutral regions predicted by tAD-seq (see Table EV3). Fourteen out of 21 predicted tADs (red) are active (enrichment > twofold above GFP; $P < 0.05$; two-sided Student's $t$-test vs. GFP control) vs. zero of eight predicted neutral regions (difference between candidate tADs and neutral regions: hypergeometric $P$-value with a pseudo-count of 1 for neutral regions).

*78* (Hr39 and Hr78) overlap the annotated ligand-binding domain, the tADs of three helix–loop–helix (HLH) TFs (HLH3B, HLH54F, and sage) overlap the annotated HLH domains (Fig 5A and B), and the tADs of five zinc finger TFs (CG10321, CG13287, CG30020, worniu, and CG30431) overlap annotated C2H2 zinc finger domains (zf-C2H2). This is interesting as both HLH and zf-C2H2 domains are typically regarded as DNA-binding domains, even though they can also mediate protein–protein interactions (Finkel *et al*, 1993; Brayer *et al*, 2008). While all three HLH-domain tADs significantly activated transcription when tested individually in luciferase assays (between 1.65-fold and 5.03-fold; $P \leq 9.8 \times 10^{-4}$; Figs 3B and 5B; Table EV3), only one of three tested zf-C2H2 tADs showed weak activity (1.52-fold; $P = 0.058$). This might be due to the disruption of the zinc fingers during the design of the test fragments (one fragment per tAD, see also below for the difficulties of defining exact tAD boundaries). Overall, these results confirm that tADs can occur at arbitrary positions within TF coding sequences and suggest that DNA-binding and trans-activation can be intrinsically linked, for example via the recruitment of transcriptional cofactors (see also e.g., Boube *et al*, 2014) or the binding of additional TFs into homo- or heterodimeric complexes.

## tADs contain regions enriched in individual amino-acid types such as glutamine, implicated in transcription modulation, protein aggregation, and phase separation

Interestingly, several tADs contain regions enriched for individual types of amino acids (Harrison, 2017; Table EV2), including histidine (H, 20 tADs), serine (S, 20 tADs), proline (P, 16 tADs), alanine (A, 10 tADs), or aspartate or glutamate (6 and 4 tADs, respectively), consistent with prior observations (Mitchell & Tjian, 1989; Gerber *et al*, 1994; Albà & Guigo, 2004; Faux *et al*, 2005; Stampfel *et al*, 2015; Hecel *et al*, 2018). Eighteen tADs, including the luciferase-validated tADs of bin, slou, Hnf4, Clock, dar1, and E2f, contain glutamine-rich (Q-rich) regions (Fig 5C), which are known to occur in transcriptional activators (Mitchell & Tjian, 1989; Gerber *et al*,

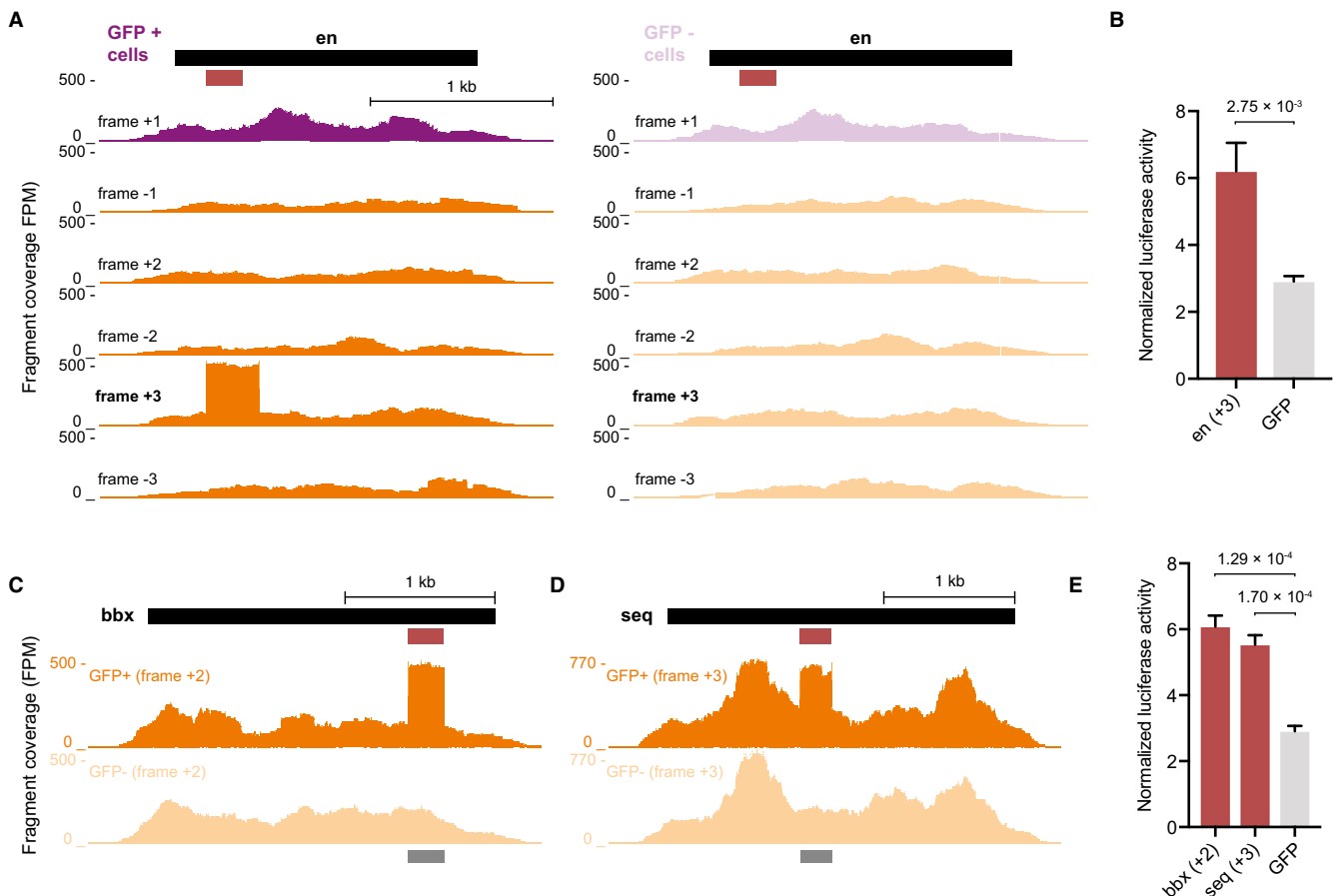

**Figure 4.  tAD-seq identifies tADs in non-native reading frames.**

A    UCSC GB screenshots displaying candidate fragment coverage from GFP+ cells (left) and GFP− cells (right) for the repressive TF engrailed (en; black bar: full-length CDS; purple: +1 frame; orange: non-native frames). The red bar indicates the region individually tested in luciferase assays. High coverage is only observed in frame +3.

B    Normalized luciferase activities (firefly/Renilla) of tAD candidate and GFP control are shown (*n* = 3, error bars: s.d., *P*-value: two-sided Student's *t*-test vs. GFP control).

C, D   Candidate fragment coverage from GFP+ and GFP− cells (orange) and the tAD calls (dark gray bar at bottom) are shown for bbx in frame +2 and seq in frame +3, respectively.

E    Normalized luciferase activities (firefly/Renilla) of bbx and seq "out-of-frame" tAD candidates and GFP control (*n* = 3, error bars: s.d., *P*-value: two-sided Student's *t*-test vs. GFP).

1994; Stampfel *et al*, 2015) and can modulate transcriptional activation (Gerber *et al*, 1994; Atanesyan *et al*, 2012; Gemayel *et al*, 2015). The presence of such Q-rich regions in short tAD sequences, including poly-Q repeats in the tADs of e.g., Clock, dar1, and E2f, emphasizes their direct involvement in transcription regulation. Given the propensity of poly-Q repeats to form protein aggregates (Halfmann *et al*, 2011; Gemayel *et al*, 2015) and liquid–liquid phase separation (Zhang *et al*, 2015), such tADs might function distinctly from other tADs, potentially via the increase in local concentrations of cofactor proteins within activating micro-environments (see below and Muerdter & Stark, 2016; Banani *et al*, 2017).

### Out-of-frame tADs contain simple sequence signatures including glutamine-rich regions

As expected (e.g., Ma & Ptashne, 1987b), the out-of-frame tADs do not contain any matches to Pfam domains, yet display several simple sequence signatures also found in TFs and in-frame tADs

(Fig 5D and Table EV4). These included regions enriched for proline (13 tADs), alanine (16), serine (11), histidine (9), and glutamine (5) (Fig 5D). In addition, 12 out-of-frame tADs have a net negative charge (Table EV4), consistent with net negative charges observed for 24 in-frame tADs (Table EV2) and the acidic tADs of different TFs [e.g., yeast Gal4 (Ma & Ptashne, 1987a) or GCN4 (Hope & Struhl, 1986)] or random sequences with tAD function (Ma & Ptashne, 1987b; Abedi *et al*, 2001). However, interestingly both in-frame and out-of-frame tADs had also net positive charges, including the validated in-frame tADs of the TFs HLH3B and sage, and the validated out-of-frame tADs of the TFs en (frame +3), Doc1 (frame +3), and bbx (frame +2).

## Results and Discussion

TFs are important transcriptional regulators and have been extensively studied by genetic and genomic means. Much is known about

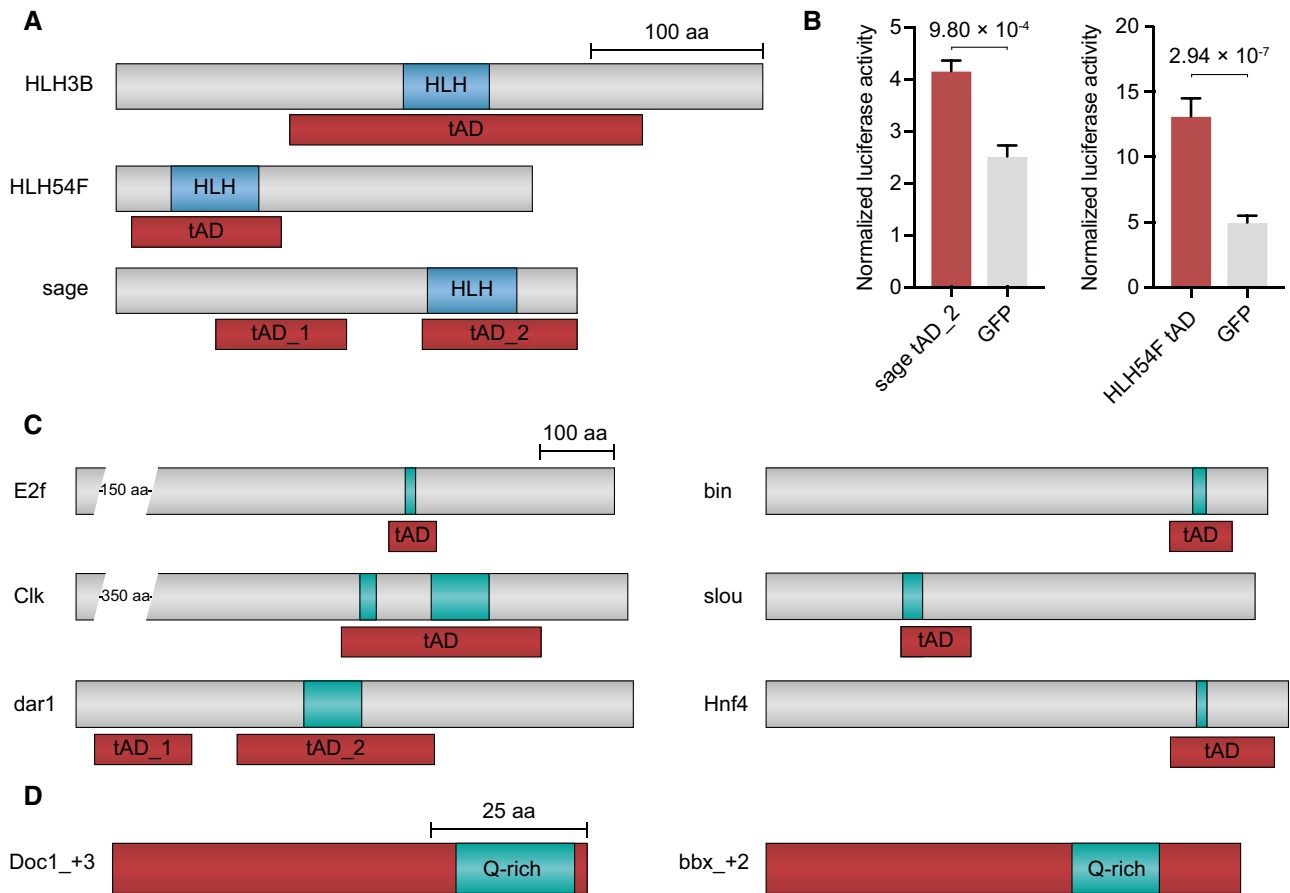

**Figure 5. Protein sequence annotation of TFs and the tADs identified by tAD-seq.**

A   The tADs of HLH3B, HLH54F, and sage overlap with basic helix–loop–helix (HLH) domains. Shown are annotated schematic views of the full-length TF CDSs (gray bars). tADs are shown in red, and the HLH domains in blue.

B   Normalized luciferase activities (firefly/Renilla) of the sage tAD_2 and HLH54F tAD candidates compared to the respective GFP control (*n* = 3 and *n* = 4, respectively, error bars: s.d., *P*-value: two-sided Student's *t*-test vs. GFP control; for HLH3B, see Fig 3B).

C   The tADs of E2f, Clk, dar1, bin, slou, and Hnf4 contain glutamine-rich (Q-rich) regions. Schematic view as in (A), and Q-rich regions in turquoise.

D   Out-of-frame tADs of Doc1 (frame +3) and bbx (frame +2) contain Q-rich regions. Displayed are the Q-rich regions (turquoise) within the tADs (red).

Data information: The scale bars in each panel apply to all elements (aa: amino acid).

TF-binding and DBD sequence properties, while tADs are less well characterized, even though they mediate trans-activation of transcription. We have developed the novel high-throughput assay tAD-seq that parallelizes prior assays to test tAD candidates individually, allowing for an unbiased identification of tADs from large candidate pools.

Applied to a large pool of candidates derived from 180 TFs of which 68 activated transcription more than twofold and 32 activated transcription more than fivefold (all others were neutral [89] or repressive [23]; Stampfel *et al*, 2015), we detected 53 tADs. Thirty-six tADs are within TFs that activate transcription more than twofold according to (Stampfel *et al*, 2015), while only five tADs were found in TFs that repress transcription more than twofold. The enrichment of tADs in activating TFs was even stronger for TFs that activate or repress more than fivefold, which contained 20 vs. zero tADs, respectively. The TFs for which we did not detect any tADs might not function with heterologous DBDs or have bi- or multi-partite tADs that consist of multiple discontinuous regions along the TF coding sequence. Such multi-partite tADs indeed exist (e.g., Herbig *et al*, 2010) and—depending on the activities of the individual constituent regions—might yield multiple detectable tADs per TF (we found four TFs with two tADs each, Table EV2) or evade detection by tAD-seq and individual candidate testing (e.g., Ma & Ptashne, 1987b; Günther *et al*, 2012). In addition, the ~ 250-bp fragments, corresponding to ~ 80 amino acids, used here should not be sufficiently long to capture long(er) tADs, which would require the testing of longer candidate fragments. Indeed, in a shallow non-exhaustive proof-of-principle screen with ~ 850-bp fragments, we found tADs not recovered when screening ~ 250-bp fragments (Tables EV5 and EV6, Appendix Figs S3 and S4). When testing nine of these, 100% (9 of 9) activated luciferase expression more than twofold and 89% (8 of 9) activated luciferase expression more than fivefold (Student's *t*-test $P < 0.05$ for all but one that had $P = 0.16$; Table EV3). For two such tADs from the taxi (tx) and CG32105 TFs, we validated that multiple short fragments spanning the long tADs were indeed not active (Fig EV1 and Table EV3), confirming that tADs can have minimal lengths below which they are not active.

As tAD-seq allows the testing of candidates of arbitrary lengths, it is possible to map the positions of long tADs with long fragments (Fig EV1) and to fine-map the locations of short tADs with shorter fragments (Fig EV2). The precise mapping of tAD boundaries seems to however remain challenging, as flanking sequences can have positive or negative effects on tAD activity: When we tested longer and shorter versions of four tADs (from MTF-1, Bteb2, HLH3B, and CG14451), we did not see a consistent trend (Fig EV3); longer and shorter versions of the same tAD could be of similar strength (MTF-1, Fig EV3A) or of different strengths with either the longer or the shorter version being stronger (Fig EV3B–D). This indicates that the precise mapping of tAD boundaries remains challenging and that tADs can have minimal lengths below which they will not function, but also that flanking regions can have a negative effect on tAD strength.

We anticipate that the sensitivity of tAD-seq depends on the complexity of the candidate library, which relates directly to the number of TFs that are assayed in parallel (here 180). Library complexity can be minimized by the use of defined, chemically synthesized DNA fragments restricted to the TFs' native (+1) reading frames, thereby reducing the number of candidates by a factor of six (for single-amino acid resolution) or—at the expense of resolution—multiples of six (note however that synthesized fragments have typically rather short maximum lengths and are fairly expensive). The use of chemically synthesized DNA fragments also allows the assessment of variant sequences with defined mutations to probe the functional importance of certain peptide motifs or individual amino acids, as has been recently done for the tAD of the yeast TF Gcn4 with a method similar to tAD-seq (Staller *et al*, 2018, published while this manuscript was under review). We further note that the strategy used here should also be applicable to equivalent screens for other protein-domain functions that can be coupled—directly or indirectly—to the expression of a selectable marker such as GFP.

The location of the identified tADs at diverse positions along the TF sequence, predominantly within unstructured regions outside Pfam domains, is consistent with known examples and explains the difficulties to computationally map tADs. Interestingly however, 12 tADs overlap structured Pfam domains including DNA-binding domains, which cautions against disregarding DBDs as tAD candidates and suggests that DNA binding and trans-activation might be intrinsically coupled, for example, by the recruitment of cofactors via the DBD (see e.g., Boube *et al*, 2014) or the homo- and heterodimerization of TFs. The different sequence signatures we observe add to the diversity of tAD sequences, which encompass also acidic- or proline-rich domains, a nine amino acid motif (9aatAD), and others (Gerber *et al*, 1994; Piskacek *et al*, 2007). Whether different peptide sequences achieve their transcription activating functions by similar or different means is an interesting open question for future research. If they function distinctly and potentially complementarily, TFs that contain different tADs might be bi-functional, similar to recent observations that the DBDs can recognize different DNA sequences (Badis *et al*, 2009).

The presence of glutamine-rich regions and extended poly-glutamine repeats in tADs is particularly exciting: Such repeats have not only been shown to modulate transcriptional activation (Gerber *et al*, 1994; Atanesyan *et al*, 2012; Gemayel *et al*, 2015) but also possess the propensity to aggregate (Halfmann *et al*, 2011; Gemayel *et al*, 2015) or promote liquid–liquid phase separation (Zhang *et al*, 2015), leading to the formation of non-membrane-bound organelles with potentially specialized micro-environments (Banani *et al*, 2017). Liquid–liquid phase separation has recently been shown to be involved in heterochromatin protein 1 (HP1)-mediated transcriptional repression and chromatin condensation (Larson *et al*, 2017; Strom *et al*, 2017) and might also play a role in transcriptional activation (Kwon *et al*, 2013; Muerdter & Stark, 2016; Hnisz *et al*, 2017). Similarly, the low-complexity domains of several gene products involved in cancer-causing translocation events can bind the C-terminal domain of Pol II and activate transcription (Kwon *et al*, 2013). It will be interesting and important to assess the role of simple sequence signatures in tADs, including glutamine-rich regions, and the potential mechanisms by which they activate transcription.

Among the interesting and important future applications of tAD-seq is the screening for tADs of TFs with distinct regulatory functions (Stampfel *et al*, 2015); for example, TFs that preferentially activate the promoters of housekeeping genes (Zabidi *et al*, 2015) or those that function exclusively in certain enhancer contexts and are therefore obligate combinatorial (Stampfel *et al*, 2015). The tADs of each of these classes of TFs are likely distinct and function via different sets of cofactors, providing unprecedented opportunities to decipher the mechanisms by which TFs and the cofactors they recruit regulate transcription in animal cells.

We anticipate that a comprehensive catalogue of tADs and regulatory peptide motifs for fly and human TFs and the detailed characterization of the protein machinery that mediates transcriptional regulation are key for our understanding of how gene expression determines development and evolution. It is also crucial at a time when enhancer function and its control by TFs and COFs are becoming increasingly central to our understanding of gene regulation in disease (Herz *et al*, 2014; Bhagwat & Vakoc, 2015) and the focus of novel therapeutic strategies (Lovén *et al*, 2013).

# Materials and Methods

### tAD-seq Gal4-DBD-candidate expression plasmid and 4xUAS-GFP reporter plasmid

We derived the tAD-seq Gal4-DBD-candidate expression plasmid (ptAD-seq-ubi63E-Gal4-DBD; Addgene ID 111930) from the fly STARR-seq plasmid (pSTARR-seq_fly; AddgeneID 71499; Arnold *et al*, 2013) by replacing sgGFP with the Gal4-DNA-binding domain (DBD) followed by a poly-glycine linker upstream of the candidate library insertion site, containing the ccdB suicide gene flanked by homology arms (used for cloning the candidates during library generation; for the library cloning strategy please see Arnold *et al*, 2013), which is followed by three stop codons (one for each reading frame). To drive the expression of the Gal4-DBD-candidate fusion proteins, we cloned the *ubiquitin-63E* promoter (from pRL-ubi63E; Addgene ID 74280) upstream of the Gal4-DBD between the KpnI and BglII sites. The 4xUAS-GFP reporter plasmid (pGL4.26_4xUAS_DSCP_GFP; Addgene ID 111930) was derived from pSGE_91_4xUAS_dCP (Stampfel *et al*, 2015; Addgene ID 71169) by replacing the luciferase gene with sgGFP (Arnold *et al*, 2013).

### Luciferase plasmids

As firefly luciferase reporter plasmid, we used pSGE_91_4xUAS_dCP (Stampfel *et al*, 2015; Addgene ID 71169) that harbors an array of 4 UAS sites upstream of the firefly luciferase gene and pRL-ubi63E (Addgene ID 74280) as Renilla control plasmid (Arnold *et al*, 2013). Candidate fragments (tADs and neutral control regions) were PCR-amplified from the respective TF Gateway entry clones (Stampfel *et al*, 2015; Table EV7). To express Gal4-DBD-candidate fusion proteins (tADs and neutral regions), we cloned them into ptAD-seq-ubi63E-Gal4-DBD (see above) using Gibson assembly between the AgeI and SalI sites. All plasmids and their full sequences are available at www.addgene.org.

### tAD-seq library generation

Gateway entry clones containing the full-length intronless TF coding (cDNA) sequences for 180 TFs of which 68 activated transcription more than twofold and 32 activated transcription more than fivefold (all others were neutral [89] or repressive [23]) were obtained from Stampfel *et al* (2015). The TF entry clones were diluted to 10 ng/μl, subsequently pooled (Table EV1), and fragmented by sonication using a Covaris S220 sonicator (10% duty cycle, eight intensity, 300 cycles per burst, 80 s for the short-, and 30 s for the long-fragment library). The sheared DNA was size-selected on a 1% agarose gel to yield 250-bp- to 350-bp-long fragments or 750-bp- to 950-bp-long fragments for the short (s)- or long (l)-fragment library, respectively (paired-end NGS confirmed that the median fragment lengths are approximately 250 and 850 bp, respectively). After gel extraction (QIAquick Gel Extraction Kit; cat. no. 28704) Illumina NEBNext Multiplexing Adaptors (NEB; cat. no. E7335 or E7500) were ligated to 1 μg of size-selected DNA fragments using NEBNext® Ultra™ II DNA Library Prep Kit for Illumina® (NEB; cat. no. E7645L) following the manufacturer's instructions, except the final PCR amplification step. Ten PCRs [98°C for 45 s, followed by 10 cycles of 98°C for 15 s, 65°C for 30 s, 72°C for 10 s/30 s (s)/(l)] with 1 μl adapter-ligated DNA as template were performed, using KAPA HiFi HotStart ReadyMix (KAPA Biosystems; cat. no. KK2602) and primers (Table EV8; fw: TTGAGCATGCACCGGACACTCTTTCCCTACACGAC GCTCTTCCGATCT and rev: ATCTATCTACGTCGA*ACTGTGGTGGA CT*AGACGTGTGCTCTTCCGATCT), which add a specific 15 nt extension to both adapters for directional cloning using recombination (Clontech In-Fusion HD; cat. no. 639650). In addition, the reverse primer renders the Illumina reverse adapter (i7) incompetent for binding of the Illumina i7 Index Read Primer (see below). Each of the five PCRs were pooled, purified, and size-selected with Agencourt AMPure XP DNA beads (ratio beads/PCR 1.4; cat. no. A63881), followed by column purification (QIAquick PCR Purification Kit; cat. no. 28106.). Cloning of the adapter-ligated, PCR-amplified candidate fragments (tAD-seq library) into the tAD-seq Gal4-DBD-candidate expression plasmid (ptAD-seq-ubi63E-Gal4-DBD) was performed by In-Fusion HD recombination as described previously (Arnold *et al*, 2013).

### Cell culture and transfection

S2 cells were cultured as described previously (Arnold *et al*, 2013). Transfection of the tAD-seq libraries and 4xUAS-GFP reporter plasmid was performed with $1.2 \times 10^9$ cells at 70–80% confluence using the MaxCyte STX Scalable Transfection System ($8 \times 10^8$ cells for the long-fragment library). Cells were transfected with 48 μg tAD-seq library plus 13.6 μg reporter plasmid per milliliter of cells at a density of $5 \times 10^8$ cells per milliliter in MaxCyte HyClone buffer mixed 1:1 with S2 culture medium without supplements using OC-400 processing assemblies (MaxCyte; cat. no. SOC-4). S2 cells were pulsed with the pre-set program "Optimization 1". Cells were transferred to a T225 cell culture flask, mixed with 10% DNase I (2,000 U/ml), and incubated for 30 min at 27°C, before resuspension in full medium. Cells were incubated post-electroporation in T225 flasks (density ~ $1 \times 10^7$ cells per ml) for 48 h before FACS and subsequent total RNA isolation of GFP$^+$ and populations.

### Flow cytometry

S2 cells were collected 48 h post-electroporation and subjected to FACS. GFP$^+$ cells (short-fragment library: replicate 1: 273,000 cells and replicate 2: 534,000 cells; and long-fragment library: replicate 1: 187,000 cells and replicate 2: 103,000 cells) were separated from GFP$^-$ cells ($1.8 \times 10^8$ GFP$^-$ cells for each of the two independent biological replicates per library) on a BD FACSAria III cell sorter, separated by a gray zone of weakly GFP$^+$ cells that barely exceeded the auto-fluorescence of un-transfected S2 cells.

## tAD-seq RNA processing

GFP$^+$ cells were mixed with $3 \times 10^6$ un-transfected S2 cells to increase the mRNA recovery of GFP$^+$ cells. Total RNA was isolated using Qiagen RNeasy Maxi or Mini Prep Kit for GFP-negative and GFP-positive cells, respectively. Poly-A+ RNA purification was performed using Invitrogen Dynabeads Oligo-dT$_{25}$ (cat. no. 610-05) followed by Ambion TURBO DNase (cat. no. AM2239) treatment according to the manufacturer's protocols, also described previously in Ref (Arnold *et al*, 2013). TURBO DNAse-treated RNA was cleaned up using Qiagen RNeasy MinElute Reaction Cleanup Kit (cat. no. 74204) according to the manufacturer's protocol. First-strand cDNA synthesis was performed with 1 μl of Invitrogen Superscript III (50°C for 60 min, 70°C for 15 min; cat. no. 18080085) using a reporter-RNA-specific primer (agttccttggcacccgagaattccaNNNNNN NNNNCGTGTGCTCTTCCGATCT) for 1–5 μg of poly-A+ RNA in 20 μl total volume. The RT primer contains 10 random nucleotides 3′ of the reverse sequencing primer binding site that we use as a unique molecular identifier (UMI) to count Gal4-DBD-candidate mRNAs (see below). Five (GFP$^-$)/two (GFP$^+$) reactions were pooled, and 1 μl of 10 mg/ml RNase A was added (37°C for 1 h) followed by bead purification (Agencourt AMPure XP DNA beads; ratio beads/RT reaction 1.8). We amplified the total amount of Gal4-DBD-candidate cDNA obtained from reverse transcription for Illumina sequencing by a 2-step nested PCR strategy using the KAPA HiFi HotStart ReadyMix (KAPA Biosystems; cat. no. KK2602). In the first PCR [98°C for 45 s, followed by 15 (GFP$^-$)/22 (GFP$^+$) cycles of 98°C for 15 s, 65°C for 30 s, 72°C for 70 s/90 s (s/l)], cDNA was amplified using 2 Gal4-candidate-specific primers (AAGCCACCATG GAAAAG*G*C*C*A*T & AGTTCCTTGGCACCCGAGAA*T*T*C), one of which spans the splice junction of the mhc16 intron (5 and 3 nucleotides at the 3′ ends are protected by phosphorothioate bonds, respectively), in a total of 10 (GFP$^-$)/2 (GFP$^+$) reactions. This specifically amplifies the Gal4-DBD-candidate cDNAs and suppresses residual plasmid background. PCR products were purified by Agencourt AMPure XP DNA beads (ratio beads/PCR 0.9) and eluted in 20 μl EB. The entire purified PCR product from each reaction served as template for the second PCR [98°C for 45 s, followed by 8–16 cycles of 98°C for 15 s, 65°C for 30 s, 72°C for 15 s/45 s (s/l)] with the KAPA Real-time Library Amplification Kit (KAPA Biosystems; cat. no. KK2702) according to the manufacturer's protocol using the following primers: i5: aatgatacggcgaccacc-gagatctacacXXXXXXXXacactctttccctacacgacgctcttccgatct (XXXXXXXX indicates the position of the index sequence for NGS; see Tables EV8 and EV9) and i7: caagcagaagacggcatacgagatGTCGTGATgtgactg-gagttccttggcacccgagaattcca, which adds the overhangs that are required for flow cell hybridization and cluster generation prior to Illumina sequencing. PCR products were purified by Agencourt AMPure XP DNA beads (ratio beads/PCR 1.4), pooled, and subjected to NGS.

## Illumina sequencing

All samples were paired-end sequenced (PE75) by the NGS unit of the Vienna Biocenter Core Facilities GmbH (VBCF) on an Illumina MiSeq system, following the manufacturer's protocol, but replacing the Illumina i7 reverse sequencing primer with a custom sequencing primer (GTGACTGGAGTTCCTTGGCACCCGAGAATTCCA).

## Luciferase reporter assays

Individual tAD candidates were tested for their ability to activate transcription by recruiting them (as Gal4-DBD fusion proteins) to a firefly luciferase reporter plasmid (via an array of four UAS sites) and measuring firefly luciferase expression. We co-transfected $1 \times 10^5$ S2 cells with 10 ng ptAD-seq-ubi63E-Gal4-DBD-candidate expression plasmids (Gal4-DBD-candidate fusion protein), 100 ng of pSGE_91_4xUAS_dCP firefly reporter plasmid (90 ng), and 10 ng of Renilla control plasmid (ubi-63E-RL) using FuGENE® HD Transfection Reagent (Promega; cat. no. E2312) according to the manufacturer's protocol. Using the Promega Dual Luciferase Assay Kit (cat. no. E1960), we measured luciferase activity at a Bio-Tek Synergy H1 fluorescence plate reader. We normalized firefly luciferase to Renilla luciferase activity. For all luciferase assays (Table EV3), we calculated standard deviations and *P*-values (two-sided Student's *t*-test) from three or four independent transfections (biological replicates) for each tAD against the GFP control.

## qPCR-based reporter assay coupled to FACS

We mixed 10 full-length TF coding sequences cloned into the pAGW-GAL4-DBD expression plasmid at an equimolar ratio (called hereafter TF mix) and transfected $5 \times 10^7$ S2 cells in three independent transfections by electroporation with 170 ng TF mix, 3,400 ng 4xUAS-GFP reporter plasmid (derived from pSGE_91_4xUAS_dCP), and 1,000 ng Renilla control plasmid (pRL-ubi63E). Forty-eight hours post-transfection, GFP$^+$ and GFP$^-$ cells were separated by FACS. GFP$^+$ and GFP$^-$ cell pools were lysed using QIAshredder columns (Qiagen; cat. no. 79654) prior to total RNA extraction using the RNeasy Mini Prep Kit (Qiagen; cat. no. 74104), with beta-Mercaptoethanol supplemented RLT buffer. 1 μg of total RNA was treated with TURBO DNase (Ambion, cat. no. AM1907) for 30 min at 37°C followed by the removal of TURBO DNase using a DNase inactivation reagent (Ambion; cat. no. AM1906). The TURBO DNase-treated RNA was reverse-transcribed using Invitrogen Super-script III and Oligo-dT$_{20}$ primers (Invitrogen; cat. no. 18418020; 50°C for 50 min, 70°C for 15 min), followed by qPCR on 2 μl diluted (1:5) cDNA using Go Tag SYBR Green qPCR Master Mix (Promega; cat. no. A6001) in a total volume of 20 μl with 0.5 μM Gal4-specific forward primer (to ensure that only cDNAs derived from transcripts originating from the Gal4-DBD-TF expression plasmids are amplified) and 0.5 μM TF-specific reverse qPCR primer (95°C, 2 min; 95°C, 3 s; 60°C, 30 s; 40 cycles total, see Table EV8 for primer sequences). Gal4-DBD-TF cDNA was normalized to Renilla luciferase cDNA, and the enrichment of normalized Gal4-DBD-TF cDNA was calculated for GFP$^+$ over GFP$^-$ cells.

## Computational analysis

### Creation of dedicated bowtie indices
Dedicated bowtie indices were made from TF and cofactor coding sequences (CDSs) used in Stampfel *et al* (2015), flanked by 2.1 kb of the upstream and downstream plasmid backbone sequence. The addition of the flanking backbone sequence allows the mapping of fragments that start or end within the plasmid backbone but include N- or C-terminal TF sequence.

 

### NGS read mapping and processing

Paired-end sequencing reads (Table EV9) were mapped to dedicated bowtie indices, created from the TFs' coding sequences, using Bowtie version 0.12.9 (Bowtie options -v 3 -m 1 –best –strata –X 2000) after removing the Y-linker and UMI sequences in the second (reverse) read. Mapped read pairs (also called "fragments") were collapsed by coordinates (start, end, strand) and by UMI, i.e., removing duplicate fragments with identical coordinates if their UMIs differed by < 3 out of the 10 nucleotides. Collapsed fragments were separated into six different reading frames based on the fragments' start coordinate and strand (as the candidate fragments were followed by a poly-A site, we did not enforce that the candidates ended in frame).

We determined the position-specific coverage for each frame using bedtools genomeCoverageBed and assessed the reproducibility between independent biological replicates by calculating the Pearson correlation coefficient between the coverage values across the CDS regions. Afterwards, we combined the fragments from both replicates for downstream analyses. We visualized tAD-seq fragment coverage using the UCSC Genome Browser with customized genome including only the TF coding sequences. We calculated enrichments, hypergeometric $P$-values, and Benjamini–Hochberg (BH)-corrected false discovery rates [FDRs; all statistical calculations done in R (Team RDC, 2008)] between the coverage values in GFP$^+$ and GFP$^-$ cells. To define tADs, we only considered regions with a minimal coverage of at least 25 independent reporter fragments (UMI) in GFP$^+$ cells and 250 fragments in GFP$^-$ cells, and selected regions with a minimal enrichment $\geq$ 1.5-fold and a hypergeometric $P$-value $\leq 1 \times 10^{-7}$ across a minimal length of $\geq$ 60 bp (20 amino acids), which we extended to include flanking coding sequences (CDS) until $P > 1 \times 10^{-3}$ over $\geq$ 60 bp (20 amino acids; tAD). We also report tADs called with a more lenient cutoff (hypergeometric $P$-value $\leq 1 \times 10^{-5}$; Tables EV2 and EV4). We assigned to each tAD the enrichment, hypergeometric $P$-value, and FDR of its summit position.

Paired-end sequencing reads for two independent biological replicates of the non-exhaustive proof-of-principal screen with a long-fragment library were processed as described above. To account for the shallow sequencing depth (~ 10-fold fewer reads), we adjusted the thresholds, considering regions with a minimal coverage of at least 10 fragments in GFP$^+$ and 25 fragments in GFP$^-$ cells for tAD calling (enrichment $\geq$ 1.5-fold and a hypergeometric $P$-value $\leq 1 \times 10^{-5}$; all other parameters and analyses were the same as above).

### TF and tAD protein sequence analysis

Protein domains were assigned using hmmscan (v. 3.1b2, Eddy, 2011) and profile hidden Markov models derived from PFAM [v. 31.0, March 2017, cite: (Finn *et al*, 2016)], with a highly significant $E$-value threshold of $1 \times 10^{-5}$. Compositionally biased regions were detected with CAST (v1.0, Promponas *et al*, 2000), segmasker (v. 1.0.0, from the NCBI blast package v. 2.6.0, Wootton & Federhen, 1996), and fLPS (Harrison, 2017). Unless specified, default settings were applied. The net charge was calculated by scoring lysine and arginine with "+1", histidine with "+0.5", and aspartic acid and glutamic acid with "−1" as in EMBOSS pepstats (v. 6.5.7.0; Rice *et al*, 2000).

### Data availability

All deep sequencing data are available at https://starklab.org and the Gene Expression Omnibus (GEO) database under the accession number GSE114387.

All read-coverage tracks, called tAD regions, enrichment tracks (GFP$^+$/ GFP$^-$), and luciferase-tested candidates/regions are available via an interactive UCSC Genome Browser session linked from https://starklab.org/data/tAD-seq_2018/.

**Expanded View** for this article is available online.

### Acknowledgements

We thank L. Cochella (IMP) for helpful comments on the manuscript and P. Heine and E. Jans (MaxCyte) for support with efficient plasmid transfection. Deep sequencing was performed at the Vienna Biocenter Core Facilities GmbH (VBCF) Next-Generation Sequencing Unit (http://vbcf.ac.at). FACS was performed at the IMP-IMBA-GMI BioOptics facility. Research in the Stark group is supported by the Austrian Science Fund (FWF, P29613-B28 and F4303-B09) and by the European Research Council (ERC) under the European Union's Horizon 2020 Research and Innovation Programme (grant agreement no. 647320). Basic research at the IMP is supported by Boehringer Ingelheim GmbH and the Austrian Research Promotion Agency [FFG, 852936 (5575353)].

### Author contributions

SW, CDA, and ASt conceived the project. SW, CDA, and FN conducted the experiments. ARW and AV performed NGS data analysis. CDA and FN analyzed data and prepared the figures. ASc analyzed the tAD protein sequences. MP and MR helped with experiments. CDA, FN, and ASt wrote the manuscript with the help of AV, partly based on text written by SW and ASt.

### Conflict of interest

The authors declare that they have no conflict of interest.

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
