## [Review Process File · The EMBO Journal]

A high-throughput method to identify trans-activation domains within transcription factor sequences

Cosmas D. Arnold, Filip Nemčko, Ashley R. Woodfin, Sebastian Wienerroither, Anna Vlasova, Alexander Schleiffer, Michaela Pagani, Martina Rath and Alexander Stark.

Review timeline:

Submission date:	21 st December 2017
Editorial Decision:	31 st January 2018
Revision received:	17 th May 2018
Editorial Decision:	12 th June 2018
Revision received:	15 th June 2018
Accepted:	15 th June 2018

Editor: Anne Nielsen

Transaction Report:

1st Editorial Decision

31st January 2018

Your study was sent out to three reviewers and we have now heard back from two of them. Since they are both supportive of your work, pending adequate revision, I have made the editorial decision at this stage to avoid further delays. Should we still receive a report from the third referee, I will forward it to you so you can consider the points for the revised study.

As you will see from the reports below, our two referees both highlight the importance and quality of the TAD-seq method and I would therefore like to invite you to submit a revised version of the manuscript, addressing their comments. I should add that it is EMBO Journal policy to allow only a single round of revision, and acceptance of your manuscript will therefore depend on the completeness of your responses in this revised version.

I generally find the reviewer comments very constructive and helpful and I would encourage you to follow them. In particular, I agree with ref #1 that the word 'TAD' itself - while being the original term used for TF trans-activating domains - has been so extensively used in HiC papers in recent years that a different name would be needed for your method.

REFeree REPORTS

Referee #1:

The manuscript by Woodfin et al addresses a very important issue, how to identify the activation domain of transcription factors (TFs). TF's DNA binding domains are generally very well structured, allowing TF's to be classified based on their DNA binding domains (e.g. Zn finger TF, bHLH, Leucine zipper etc). In contrast, TF's activation domains - the part of the transcription that actually does the business of activating or repressing transcription - are very poorly characterized. Actually

nothing has progressed in this area since the late 80's and early 90's. All we know is that they are generally unstructured, acidic domains. As such, they cannot be readily identified computationally. Given that, any method that can functionally identify activation domains is important, and it is for this reason that I am very positive about this paper.

The experiments are well thought through, and the data is of very high quality, in keeping with the quality of this lab. There are a couple of things I suggest for improvement, which are mainly in way the data is discussed

1) While it makes perfect sense to use the abbreviation of TAD-seq for trans-activation domains, given the now rampant use and discussion of Hi-C TADs as Topologically Associated Domains, I suggest that the authors use another abbreviation - either TD-seq or AD-seq (Activation Domains, which is my personal favorite). This will really help the reader - I had to convert my brain from Hi-C TADs to activation domains throughout the entire paper. Plus, the transcription field has enough complexity in its nomenclature, which I'm sure the authors appreciate.

2) One of the most surprising findings for me in this manuscript is the fact that only ~10% of TFs (19 out of the 180 tested) could activate (or repress) transcription in this assay. This is important information and I'd like to see a lot more discussion about this in the manuscript. First, for the 19 that could function - where these among the top activators and/or repressors in the Stampfel et al study? Do they have anything in common, either the type of DNA binding domain, or activation domain (acidic, globular etc) Second, for the ~160 or so factors that couldn't function in this study: How many of those gave activity in Stampfel et al? For the ones that did, this clearly indicates that it is not missing co-factors etc in S2 cells that is the reason for their lack of AD activity here. These factors AD may be activated by a conformation change induced by DNA binding that is not recapitulated when Gal4 binds to UAS sequences. The authors should discuss this possibility. For the TFs that didn't activate in Stampfel et al, did the authors try another cell line? Have these factors anything in common. Are they activated by signaling pathways etc.?

I see this large pool of 'negative results' as more evidence for how much of a big unsolved puzzle TF activation domains are. We have made very little progress in this area in the last 30 years. The identification of multiple ADs for some TFs (presumably 3 of the 19) should also be discussed. This fits with data from Martha Bulyk and others showing that some TFs have two sequence specific DNA binding motifs.

3) Page 4 - the authors selected 180 TFs, 68 activated transcription >2 fold, with 32 activating transcription >5 fold. What about the other 112 TFs? Are they repressors? Did they activate transcription <2-fold in Stampfel et al?

Minor

Page 4 - GFP-FACS can enrich for transcription activating factors section.

Sentence, "As expected, the two strongest activators were enriched in GFP+ cells, while the strongest repressors were most strongly enriched in GFP- cells". To make things clearer, it would be good to put the names of the two strongest and weakest into the sentence.

"As expected, the two strongest activators (sox14, lab) were enriched in GFP+ cells, while the strongest repressors (gro, mirr) were most strongly enriched in GFP- cells".

Referee #2:

This manuscript describes a high-throughput method, TAD-seq, for finding transcription activation domains (TADs) within transcription factors. The authors generate fusion proteins between the DNA-binding domain of Gal4 and a library of random genomic fragment (in-frame and out-of-frame) from the coding regions of 180 known fly transcription factors. They put these fusion gene constructs under the control of a constitutive promoter and transfected them on plasmids into fruit fly S2 cells, which were also transfected with a second plasmid. This second plasmid contained a GFP reporter gene under the control of 4 UAS to which the fusion protein can bind with its Gal4-DBD. Those genomic fragments that were translated into functional activation motifs activated the

transcription of GFP. Subsequently, the S2 cells were FACS sorted and the random fragment regions of GFP+ and GFP- cells were sequenced separately using paired-end sequencing. Paired reads were mapped to the genome and a coverage for GFP+ and GFP- libraries along the 6 frames of each TF was computed. A significant log enrichment of coverage in GFP+ over GFP- libraries indicated a functional activation motif.

This manuscript addresses an important question: what properties make a sequence activate transcription. Although this question has been studied for a long time, new insights have been relatively few and far between. This seems to be changing with the application of high-throughput methods to this question. The manuscript presents a new HT method and finds 21 activation regions in 19 TFs out of 180 TFs tested. Interestingly, they find several TADs overlapping Zn finger motifs and HLH "DNA-binding" domains.

The manuscript is very well written and gives a good background of the field.

Major points:

1. It is difficult to get a feeling for how reliable the method is as only a tiny fraction of the data is shown. I urge the authors to show in a supplemental figure at least the equivalent of Figure 2 for all 19 TFs for which TADs were found, and also for a few *randomly selected* TFs without detected TADs. Also, position indices should be given to be able to estimate the position of the peaks in the TF sequences.

I wonder, for instance, how good the resolution of the method is. With a fragment length of 100 amino acids, I would expect it to be quite low. Can the resolution be demonstrated (visually) at the example of a TAD with a short, well-defined activation motif?

2. To allow the community to build on these results, all coverage data from GFP+ and GFP- library reads mapped to the TFs in all 6 frames should be available in a supplemental file, as well as the annotations described in Methods section "TF and TAD amino acid/protein sequence analysis". Raw sequencing data should be uploaded to a public sequencing archive.

3. Ma and Ptashne (Cell 1987) used a similar assay to find TADs. Instead of fusing fragments from coding sequences of TFs downstream of a DBD gene, they took genomic fragments from *E. coli*. They obtained 154 blue colonies among about 15000 transformants, a rate of 1%. Given that the GC content of the *E. coli* and *D. melanogaster* genomes is similar and given that the length of the *E. coli* genomic fragments was rather shorter than the 300 bp used in the present study, I would expect to see a fraction $\geq 1\%$ of sequences to be activatory. Assuming that the average length of the coding regions of the 180 TFs is 1000bp and the fragments are length 300bp, I would expect $\geq 1\% * 180 * (1000 \text{ bp} / 300\text{bp}) * 5 \text{ (frames)} = 30$ activatory sequences that are translated out of frame, whereas the authors observe none. It would be important to address the causes of this discrepancy.

Could it be due to a too conservative, stringent choice of FDR threshold used by the authors? How were the thresholds on the P-value ($1.3E-7$) and on the FDR ($1.2E-6$) chosen? In particular the latter looks exceedingly strict.

I would encourage the authors to analyse predictions at lower FDRs to gain insights about what general properties can make a sequence activatory.

Minor points:

5. The following, mostly overlooked study should be cited and discussed, as they have performed a high-throughput screen of transactivating random peptides and obtained interesting results: Abedi, M. et al. Transcriptional Transactivation by Selected Short Random Peptides Attached to Lexa-Gfp Fusion Proteins. *BMC Mol Biol* 2, 10 (2001).

6. For how many out of the 180 TFs are the predicted disordered regions longer than 40 residues split between two or more exons? Is the set of TFs without disordered regions split between exons enriched among the TFs with detected TADs? In other words, could the presence of introns

disrupting the coding regions of some potential TAD regions explain why the TADs could not be detected with TAD-seq? Could this potential source of loss of sensitivity be addressed by using cDNA reverse-transcribed from TF mRNAs to generate the genomic fragment library?

1st Revision - authors' response

17th May 2018

Referee #1:

The manuscript by Woodfin et al addresses a very important issue, how to identify the activation domain of transcription factors (TFs). TF's DNA binding domains are generally very well structured, allowing TF's to be classified based on their DNA binding domains (e.g. Zn finger TF, bHLH, Leucine zipper etc). In contrast, TF's activation domains - the part of the transcription that actually does the business of activating or repressing transcription - are very poorly characterized. Actually nothing has progressed in this area since the late 80's and early 90's. All we know is that they are generally unstructured, acidic domains. As such, they cannot be readily identified computationally. Given that, any method that can functionally identify activation domains is important, and it is for this reason that I am very positive about this paper.

The experiments are well thought through, and the data is of very high quality, in keeping with the quality of this lab. There are a couple of things I suggest for improvement, which are mainly in way the data is discussed.

1) While it makes perfect sense to use the abbreviation of TAD-seq for trans-activation domains, given the now rampant use and discussion of Hi-C TADs as Topologically Associated Domains, I suggest that the authors use another abbreviation - either TD-seq or AD-seq (Activation Domains, which is my personal favorite). This will really help the reader - I had to convert my brain from Hi-C TADs to activation domains throughout the entire paper. Plus, the transcription field has enough complexity in its nomenclature, which I'm sure the authors appreciate.

We fully agree that the terminology needs to avoid ambiguity with the currently widely used Hi-C TADs. We now consistently use tAD to abbreviate trans-activation domains and tAD-seq throughout the manuscript and explain this choice with respect to Hi-C TADs on page 2. We hope this is sufficiently distinct (AD-seq would also work but we prefer tAD-seq given the originally established nomenclature).

2) One of the most surprising findings for me in this manuscript is the fact that only ~10% of TFs (19 out of the 180 tested) could activate (or repress) transcription in this assay. This is important information and I'd like to see a lot more discussion about this in the manuscript. First, for the 19 that could function - where these among the top activators and/or

repressors in the Stampfel et al study? Do they have anything in common, either the type of DNA binding domain, or activation domain (acidic, globular etc)

Many thanks for pointing out that we had not sufficiently clearly explained the nature of the 180 TF library and the tADs we uncovered. Of the 180 TFs, only 68 activated transcription >2-fold and 32 activated transcription >5-fold, while all others were neutral (89) or repressive (23), and – as the reviewer suspects – the identified tADs were predominantly found in the more strongly activating TFs (or the originally reported tADs, 14 were within TFs that activate >2-fold and 9 within TFs that activate >5-fold, while no tAD was found in TFs that repressed transcription >2 or >5-fold, respectively).

We further explored the influence of two critical parameters on the number of tADs in order to make tAD-seq more sensitive, sequencing depth and candidate fragment length. Realizing that our initial sequencing might not have been sufficiently exhaustive, we have now sequenced the previous libraries approx. 10-times more deeply, which uncovers 53 tADs (66 tADs with more lenient cutoffs). Given the increased sensitivity and tAD number, we have also extended the luciferase validations (Table EV1 and EV3). We have also performed a non-exhaustive proof-of-principle screen with ~850bp fragments, more than three times the length of the original screen (~250bp). This screen indeed recovers tADs that were not found when using shorter fragments and we demonstrate for two such tADs that none of several short fragments spanning the long tADs function in luciferase assays. We have included the results from the 850bp pilot screen as new Tables EV5 and EV6 and compare the two screens in two new Expanded View Figures EV1 and EV2.

Overall, we have extended the luciferase validations (Tables EV2 – EV4), updated the manuscript throughout to reflect the improved sequencing depth, and added the screen with 850bp long fragments (Tables EV5 and EV6, Extended View Figures EV1 & EV2). We have also extended the discussion about other possible factors that can influence the detection of tADs.

Second, for the ~160 or so factors that couldn't function in this study: How many of those gave activity in Stampfel et al? For the ones that did, this clearly indicates that it is not missing co-factors etc in S2 cells that is the reason for their lack of AD activity here. These factors AD may be activated by a conformation change induced by DNA binding that is not recapitulated when Gal4 binds to UAS sequences. The authors should discuss this possibility.

We are sorry for not sufficiently clearly explaining the nature of the 180 TF library (see above): only 68 activated transcription >2-fold and 32 activated transcription >5-fold, while all others were neutral (89) or repressive (23). We have now increased the sensitivity of the assay (see above), but additionally discuss the possibility that some tADs might not be compatible with heterologous DBS (see discussion on page 8).

For the TFs that didn't activate in Stampfel et al, did the authors try another cell line?

Have these factors anything in common. Are they activated by signaling pathways etc.?

We did not try another cell line, also because tADs are reported to function similarly across cell lines (e.g. Stampfel et al 2015), even across different species (e.g. Fischer et al., 1988, Kakidani et al., 1988 or Ma et al., 1988). However, we agree that testing tAD activity in different cell lines and under different signaling conditions will be an interesting addition in the future.

I see this large pool of 'negative results' as more evidence for how much of a big unsolved puzzle TF activation domains are. We have made very little progress in this area in the last 30 years. The identification of multiple ADs for some TFs (presumably 3 of the 19) should also be discussed. This fits with data from Martha Bulyk and others showing that some TFs have two sequence specific DNA binding motifs.

We agree that the last 30 years have seen little progress in our understanding of tADs and how they function. We hope that this work will provide a new starting point for the identification of tADs and their molecular functions. The observation that 4 TFs have 2 tADs each is indeed important and we now discuss these TFs and the analogy to Martha Bulyk's data that TFs recognize distinct DNA sequence motifs on pages 8 and 10, citing the Badis et al 2009 reference. Many thanks!

3) Page 4 - the authors selected 180 TFs, 68 activated transcription >2 fold, with 32 activating transcription >5 fold. What about the other 112 TFs? Are they repressors? Did they activate transcription <2-fold in Stampfel et al?

All others were indeed neutral (89) or repressive (23), we now mention this more prominently on pages 4, 8, and 12.

Minor

Page 4 - GFP-FACS can enrich for transcription activating factors section.

Sentence, "As expected, the two strongest activators were enriched in GFP+ cells, while the strongest repressors were most strongly enriched in GFP- cells". To make things clearer, it would be good to put the names of the two strongest and weakest into the sentence.

"As expected, the two strongest activators (sox14, lab) were enriched in GFP+ cells, while the strongest repressors (gro, mirr) were most strongly enriched in GFP- cells".

We added the names as suggested, many thanks.

Referee #2:

This manuscript describes a high-throughput method, TAD-seq, for finding transcription activation domains (TADs) within transcription factors. The authors generate fusion proteins between the DNA-binding domain of Gal4 and a library of random genomic fragment (in-frame and out-of-frame) from the coding regions of 180 known fly transcription factors. They put these fusion gene constructs under the control of a constitutive promoter and transfected them on plasmids into fruit fly S2 cells, which were also transfected with a second plasmid. This second plasmid contained a GFP reporter gene under the control of 4 UAS to which the fusion protein can bind with its Gal4-DBD. Those genomic fragments that were translated into functional activation motifs activated the transcription of GFP. Subsequently, the S2 cells were FACS sorted and the random fragment regions of GFP+ and GFP- cells were sequenced separately using paired-end sequencing. Paired reads were mapped to the genome and a coverage for GFP+ and GFP- libraries along the 6 frames of each TF was computed. A significant log enrichment of coverage in GFP+ over GFP- libraries indicated a functional activation motif.

This manuscript addresses an important question: what properties make a sequence activate transcription. Although this question has been studied for a long time, new insights have been relatively few and far between. This seems to be changing with the application of high-throughput methods to this question. The manuscript presents a new HT method and finds 21 activation regions in 19 TFs out of 180 TFs tested. Interestingly, they find several TADs overlapping Zn finger motifs and HLH "DNA-binding" domains.

The manuscript is very well written and gives a good background of the field.

Major points:

1. It is difficult to get a feeling for how reliable the method is as only a tiny fraction of the data is shown. I urge the authors to show in a supplemental figure at least the equivalent of Figure 2 for all 19 TFs for which TADs were found, and also for a few *randomly selected* TFs without detected TADs. Also, position indices should be given to be able to estimate the position of the peaks in the TF sequences.

We agree that readers should be able to explore the entire dataset and get precise information on the positions of the peaks in the TF sequences. As suggested, we therefore now provide the equivalent of Fig. 2 for all TFs for which tADs were found (Appendix Figures S1-S4). In addition, we provide three publicly accessible UCSC genome-browser session that allows the interactive inspection of the raw data (coverage profiles, enrichments, and tAD calls) linked from https://starklab.org/data/tAD-seq_2018/. In detail, we provide one session for frame +1 for short- and long-fragment tAD-seq (<https://goo.gl/BAcLnX>), one

session for short-fragment tAD-seq (all 6 frames) (<https://goo.gl/RCwzmm>) and one session for long-fragment tAD-seq (all 6 frames) (<https://goo.gl/ebGgwz>). We further provide all tAD coordinates, protein sequences, protein analyses results, and luciferase results in the form of Expanded View Tables EV2, 4, 6, 7. Finally, we uploaded all raw and processed NGS data to GEO (accession number GSE114387).

I wonder, for instance, how good the resolution of the method is. With a fragment length of 100 amino acids, I would expect it to be quite low. Can the resolution be demonstrated (visually) at the example of a TAD with a short, well-defined activation motif?

We agree that the resolution of the method will relate to the fragment-length used, and screening longer or shorter fragments have different advantages. We now performed a non-exhaustive proof-of-principle screen with fragments of ~850bp (vs. ~250bp used for the original screen; Expanded View Figs EV1-EV3; Tables EV5 and 6). This demonstrates that longer fragments allow the detection of longer tADs for which shorter fragments are inactive (Expanded View Figs EV1-EV3). In contrast, for short tADs screening with shorter fragments can determine the location of the tAD more precisely (Expanded View Fig EV2). We further tested slightly longer and shorter variants of 4 tADs and found an inconsistent trend with the longer fragments being stronger, or weaker, or of the same strength (Expanded View Fig EV3). This indicates that tADs might not have defined boundaries and the precise delineation of tADs will depend on specific thresholds.

Overall, these additional datasets show that a detailed mapping of tAD boundaries at high resolution will require screens at several different fragment lengths, with a tradeoff between resolution and sensitivity with respect to long tADs (see our response to reviewer 1 comment 2 above). We now add these new datasets (Expanded View Figs EV1-EV3; Tables EV5 and 6) and discuss how screens with different fragment lengths can be used to map longer tADs or fine-map the location and boundaries of shorter tADs (pages 8-9).

2. To allow the community to build on these results, all coverage data from GFP+ and GFP- library reads mapped to the TFs in all 6 frames should be available in a supplemental file, as well as the annotations described in Methods section "TF and TAD amino acid/protein sequence analysis". Raw sequencing data should be uploaded to a public sequencing archive.

We agree and now provide the equivalent of Fig. 2 for all TFs for which tADs were found (Appendix Figures S1-S4), 3 publicly accessible UCSC genome-browser session (permanently linked from https://starklab.org/data/tAD-seq_2018/), and uploaded all raw and processed NGS data to GEO (accession number GSE114387).

3. Ma and Ptashne (Cell 1987) used a similar assay to find TADs. Instead of fusing

fragments from coding sequences of TFs downstream of a DBD gene, they took genomic fragments from *E. coli*. They obtained 154 blue colonies among about 15000 transformants, a rate of 1%. Given that the GC content of the *E. coli* and *D. melanogaster* genomes is similar and given that the length of the *E. coli* genomic fragments was rather shorter than the 300 bp used in the present study, I would expect to see a fraction $\geq 1\%$ of sequences to be activatory. Assuming that the average length of the coding regions of the 180 TFs is 1000bp and the fragments are length 300bp, I would expect $\geq 1\% * 180 * (1000 \text{ bp} / 300\text{bp}) * 5 \text{ (frames)} = 30$ activatory sequences that are translated out of frame, whereas the authors observe none. It would be important to address the causes of this discrepancy.

Could it be due to a too conservative, stringent choice of FDR threshold used by the authors? How were the thresholds on the P-value ($1.3E-7$) and on the FDR ($1.2E-6$) chosen? In particular the latter looks exceedingly strict. I would encourage the authors to analyse predictions at lower FDRs to gain insights about what general properties can make a sequence activatory.

We agree that the results of Ma & Ptashne (Cell 1987) argues that some of the out-of-frame fragments should be active. While we had inspected all frames for the positive control MTF-1 (Fig. 2), we only analyzed reading frame +1 during tAD calling across all TFs. This omission rather than potentially overly stringent thresholds explains the absence of such tADs. Many thanks for pointing out that this has been unclear.

We agree with the reviewer that an analysis of the non-native frames would be highly interesting and have now performed this analysis. The non-native frames indeed contain tADs that validate in luciferase assays (new Fig 4) and contain simple sequence signatures such as glutamine-rich regions (but no Pfam domains) also seen in in-frame tADs. We added the identification and validation of out-of-frame tADs (new Fig 4) and the analyses of their protein sequence properties (new Fig 5D and Table EV4). In addition, we made tAD-seq more sensitive, uncovering 53 tADs (66 tADs with more lenient cutoffs; see our response to reviewer 1 above). Given the increased sensitivity and the additional tADs predicted, we have also extended the luciferase validations (Fig 3 and Tables EV2 and EV3). Many thanks for these important suggestions.

Minor points:

5. The following, mostly overlooked study should be cited and discussed, as they have performed a high-throughput screen of transactivating random peptides and obtained interesting results:

Abedi, M. et al. Transcriptional Transactivation by Selected Short Random Peptides Attached to Lexa-Gfp Fusion Proteins. BMC Mol Biol 2, 10 (2001).

We now cite this study throughout the manuscript, thanks.

6. For how many out of the 180 TFs are the predicted disordered regions longer

than 40 residues split between two or more exons? Is the set of TFs without disordered regions split between exons enriched among the TFs with detected TADs? In other words, could the presence of introns disrupting the coding regions of some potential TAD regions explain why the TADs could not be detected with TAD-seq? Could this potential source of loss of sensitivity be addressed by using cDNA reverse-transcribed from TF mRNAs to generate the genomic fragment library?

The fragment library was constructed from cDNA, such that the disruption of coding regions by introns is not an issue. We have now revised the main text (page 4) and methods section ('TAD-seq library generation', page 12) to improve clarity, thanks for pointing out that this has not been sufficiently clear.

2nd Editorial Decision

12th June 2018

Thank you for submitting a revised version of your manuscript. It has now been seen by the two original referees and their comments are included below.

As you will see, the referees both find that all criticisms have been sufficiently addressed and recommend the manuscript for publication. However, before we can go on to officially accept the manuscript there are a few editorial issues concerning text and figures that I need you to address in final revision.

REFeree REPORTS

Referee #1:

The resubmitted paper is much improved, including the more extensive discussion. The authors have addressed all of my concerns. I therefore recommend publication

Referee #2:

All issues raised by the two reviewers have been extensively and constructively addressed in my view. I thank the authors for these improvements.

Corresponding Author Name: Alexander Stark

Manuscript Number: EMBOJ-2017-98896